# Effectiveness of Palliative Care before Death in Reducing Emergency Care Utilization for Patients with Terminal Cancer and Trends in the Utilization of Palliative Care from 2005–2018

**DOI:** 10.3390/healthcare11212907

**Published:** 2023-11-06

**Authors:** Yi-Shiun Tsai, Wen-Chen Tsai, Li-Ting Chiu, Pei-Tseng Kung

**Affiliations:** 1Department of Orthopedics, Feng Yuan Hospital, Feng Yuan, Taichung 420210, Taiwan; stcomas@gmail.com; 2Department of Health Services Administration, College of Public Health, China Medical University, Taichung 406040, Taiwan; wtsai@mail.cmu.edu.tw (W.-C.T.); litingchiu933@gmail.com (L.-T.C.); 3Department of Healthcare Administration, Asia University, Taichung 413305, Taiwan; 4Department of Medical Research, China Medical University Hospital, Taichung 404327, Taiwan

**Keywords:** terminal cancer patients, palliative care, emergency care utilization, CPR (cardiopulmonary resuscitation), endotracheal intubation, ICU (intensive care unit) admission

## Abstract

This retrospective cohort study aimed to examine the effect of palliative care for patients with terminal cancer on healthcare utilization. The National Health Insurance (NHI) Research Database and death certificates were utilized to identify patients who died of cancer between 2005 and 2018. The number of terminal cancer patients between 2005 and 2018 was 605,126. Propensity score matching and conditional logistic regression were performed. The odds ratios (ORs) for “emergency care utilization”, “CPR”, “endotracheal intubation”, and “ICU admission” were significantly lower for enrolled patients regardless of enrollment time compared to unenrolled patients. Compared to unenrolled patients, the OR for “emergency care utilization” increased from 0.34 to 0.68, the OR for “CPR use” increased from 0.13 to 0.26, the OR for “intubation” increased from 0.15 to 0.26, and the OR for “ICU admission” increased from 0.27 to 0.40 in enrolled patients. Between 2005 and 2010, CPR utilization, intubation, and ICU admission in patients enrolled in palliative care declined each year. Since the inclusion of palliative care in NHI (from 2010 onward), its utilization has increased slightly each year. Patients with terminal cancer enrolled in palliative care consume fewer medical resources before death than unenrolled patients; however, the difference decreases with longer times before death.

## 1. Introduction

Cancer has a high incidence and mortality rate, with 20 million new cases worldwide [1] and nearly 10 million deaths in 2020 [2]. With advances in medical treatment, the overall survival of patients with cancer has increased, treatment and care have become complex, medical costs have increased, and the quality of care for cancer patients has become a major healthcare issue [3]. The disease trajectory of patients with cancer is more predictable than those with other diseases [4,5]. Palliative care has gradually gained acceptance as a means to reduce the discomfort of patients with terminal cancer that are suffering from physical and psychological distress owing to the disease metastasizing and becoming incurable [6]. These patients seek emergency care, resulting in higher utilization of emergency care resources [7]. Studies have shown that 77% of patients with terminal cancer have had at least one emergency room visit in the last 3 months of life [8]. However, some of these visits could have been avoided [9]; therefore, emergency care utilization at the end of life is considered an important indicator of poor-quality end-of-life cancer care [10,11]. For death with dignity, the use of ineffective medications that increase the patient’s pain, as well as life-sustaining medical treatments, including intensive care unit (ICU) admission, cardiopulmonary resuscitation (CPR), and endotracheal intubation, is avoided. Extensive use of futile therapies in patients with terminal cancer is an indicator of inappropriate palliative care [12,13]. In Taiwan, 10–30% of cancer patients utilize emergency and intensive care services and undergo emergency medical procedures within 1 month before death. On average, 24.3% of cancer patients receive intubation therapy, and 10.6% undergo CPR (cardiopulmonary resuscitation) [14]. This tendency may be attributed to the discomfort experienced by patients and the subsequent emergency hospital visits. Family members, due to a lack of understanding of the terminal condition of the patient, inadequate caregiving capacity, unpreparedness for impending death, or absence of a DNR (Do Not Resuscitate) order, often resort to emergency services. In cases where patients exhibit life-threatening conditions, emergency physicians, possibly due to the absence of a DNR order from the patient or under the insistence of family members, may engage in aggressive life-saving measures (CPR; patients are often also intubated) and eventually admit the patient to an ICU (intensive care unit).

Hospice and palliative care is a care model that helps patients manage their symptoms through accurate assessment, reduces the spiritual suffering of the patient and their family members, and emphasizes improving the quality of life of the patient. Palliative care can control symptoms such as shortness of breath and pain in patients with terminal cancer [15] and even prolong survival despite involving less aggressive treatment [16]. Such patients enrolled in hospice and palliative care had lower rates of emergency care utilization compared to unenrolled patients [17,18]. Hospice and palliative care can also reduce the relative probability of ICU admission and length of stay [19]. In addition, Kao et al. compared the differences in futile life-sustaining measures such as CPR and intubation between cancer patients that were and were not enrolled in hospice and palliative care and found that cancer patients enrolled in hospice and palliative care were less likely to receive CPR [20]. Wang et al. showed that terminal pancreatic cancer patients enrolled in hospice and palliative care were less likely to be intubated [21]. Over the past few years (2001–2018), the rate of terminal cancer patients in Taiwan enrolled in hospice and palliative care 1 year before death has increased annually from 7% in 2000 to 62.8% in 2018 [14,22]. Hospice and palliative care interventions can reduce indicators of poor terminal cancer care, including emergency care utilization, ICU admission, CPR, and intubation [23,24].

Many previous studies have focused on the impact of hospice and palliative care on the quality of life of patients with terminal cancer; however, few studies were conducted on the relationship between “duration of palliative care” and “consumption of healthcare resources.” Previous studies on the duration of palliative care have analyzed emergency care utilization and ICU admission 1–6 months before death and have shown that palliative care at early stages is relatively less costly and requires less healthcare resource utilization [25,26]. The American Society of Clinical Oncology (ASCO) recommends palliative care within 8 weeks of a terminal cancer diagnosis [27]. The present study was conducted to examine the effect of the “duration of palliative care” on emergency care utilization, CPR, ICU admission, and intubation in patients with terminal cancer before death from 2005 to 2018. The primary study objective was to compare differences in emergency care utilization, CPR, intubation, and ICU admission among patients with terminal cancer (enrolled and not enrolled in palliative care) at 1, 4, 8, 12, and 24 weeks before death. The secondary objective was to understand trends in the utilization of palliative care by these patients before death from 2005 to 2018.

## 2. Materials and Methods

### 2.1. Data Sources and Participants

This was a retrospective cohort study that utilized data from the National Health Insurance (NHI) Research Database maintained by Taiwan’s Ministry of Health and Welfare from 2005 to 2018 in addition to data on cause of death. Deceased patients whose primary and secondary causes of death were cancer were included. Cancer is defined as the first three characters for categories 140–208 in the ICD-9-CM codes and the first three characters for categories C00-C97 in the ICD-10-CM codes. There were 609,978 deaths diagnosed as cancer (D-code: 140–208, ICD10: C00-C97) between 2005 and 2018. After excluding 11,503 cases with incomplete data on age (3566), marital status (206), monthly salary (3241), education level (1249), and urbanization level of resident area (3241), 605,126 cases were included in the study. Figure 1 presents a flowchart of the selection process.

The data in the study were de-identified, and the study was conducted in accordance with the principles of the Declaration of Helsinki and was approved by the Institutional Review Board of Jen-Ai Dali Hospital (IRB no.: JAH IRB 110-38).

### 2.2. Variables Description

In this study, “enrollment in palliative care” was determined as the independent variable; it was defined as enrollment in “hospice and palliative care” with a palliative-care-related payment system and standardized reporting code, providing inpatient care, outpatient services, and/or home care. “Emergency care utilization” was defined as patients reporting with an emergency during the observation period. Emergency CPR, intubation, and ICU admission were defined as the presence/absence of reported use of CPR (47029C), the presence/absence of endotracheal intubation (47031C), and the presence/absence of ICU admission in the study population (03010E, 03011F, 03012G, 03013H, 03010E, 03011F, 03012G, 03013H, 02011K, 02012A, 0013B). The remaining control variables were as follows: (i) sex (male or female); (ii) age (<55 years, 55–64 years, 65–74 years, 75–84 years, and ≥85 years); (iii) educational status (illiterate or below elementary-school level, middle school, high school [vocational school], junior college, and university and above); (iv) marital status (unmarried, married, divorced, or widowed). Monthly salary was classified into 7 levels (in New Taiwan dollars [TWD]): ≤17,280, 17,281–22,800, 22,801–28,800, 28,801–36,300, 36,301–45,800, 45,801–57,800, and ≥57,801. The urbanization level of the towns and villages in Taiwan was classified using a 1–7 scale, with 1 indicating the most urbanization and 7 indicating the least urbanization [28]. The Charlson Comorbidity Index (CCI) was used to represent the health status of the patients in the study population based on medical diagnostic data collected in the 2 years prior to patients’ deaths. Primary and secondary diagnostic data were converted into the CCI based on the Deyo–Charlson Comorbidity Index. Because the study population consisted of cancer patients, the cancer score was not considered for the calculation of the severity of comorbidities. The CCI was classified as 0, 1, 2, or ≥3 points, with higher values representing more severe comorbidity [29]. Cancer cases were classified according to primary cause of death as lung, liver, colorectal, breast, oral, prostate, gastric, pancreatic, esophageal, cervical, other, or multiple cancers. Regarding hospital levels and attributes, the medical institution in which a patient with terminal cancer was first enrolled in palliative care was the primary medical institution. For patients not enrolled in palliative care, records of cancer-related medical visits in the 3 months before death were examined, and the medical institution with the most visits was defined as the primary medical institution. Major medical institutions were further sub-classified into medical centers, regional hospitals, district hospitals, and primary clinics. Healthcare providers were classified as public or private.

### 2.3. Statistical Analysis

The SAS 9.4 software (SAS Institute Inc., Cary, NC, USA) was used for statistical analysis of the NHIRD data. Two-tailed *p* values of <0.05 were considered statistically significant.

First, descriptive statistics were used to show the utilization and trends of palliative care for patients with terminal cancer by year (2005–2018). Patients with terminal cancer were divided into five groups based on enrollment in palliative care for 1, 4, 8, 12, or 24 weeks before death. Differences in the utilization of emergency and critical care resources were compared between the enrolled and non-enrolled groups before death. Propensity score matching (PSM) was performed to reduce the selection bias between enrolled and non-enrolled groups. Logistic regression analysis was performed with “enrollment in palliative care” as the dependent variable and sex, age, severity of comorbidities, and cancer type as independent variables. After estimating probabilities (i.e., propensity scores from 0–1), greedy nearest neighbor matching with the caliper method was used for 1:1 matching. Consequently, patients were classified into two groups: “enrolled in palliative care” and “not enrolled in palliative care”.

Bivariate analysis was performed using the chi-square test to examine whether there were significant differences in emergency care utilization, CPR, ICU, and intubation between those enrolled in palliative care at 1, 4, 8, 12, and 24 weeks before death and those not enrolled. Next, we examined the effects of enrollment in palliative care at 1, 4, 8, 12, and 24 weeks before death on emergency care utilization, CPR, ICU admission, and intubation; these were the dependent variables, and “enrollment in palliative care at 1, 4, 8, 12, or 24 weeks before death” was the independent variable. The control variables included baseline characteristics, economic factors, environmental factors, health status, cancer type, characteristics of the primary medical institution, and the year of death. Conditional logistic regression was conducted to examine the effects of enrollment in palliative care on emergency care utilization, CPR, ICU admission, and intubation of terminal cancer patients. We compared the ratios of the percentages of emergency, CPR, ICU, and endotracheal intubation utilization rates (enrollment utilization divided by non-enrollment utilization) of those who were enrolled or not enrolled in palliative care 1, 4, 8, 12, and 24 weeks before death, respectively, to observe the trends in the ratio of the utilization rate of each category during the years of death (2005–2018), and compared them with the graphs.

## 3. Results

### 3.1. Utilization and Trends of Palliative Care for Patients with Terminal Cancer before Death

Between 2005 and 2018, 609,978 cancer-related deaths were recorded. After excluding cases with incomplete data on monthly salary, educational attainment, and medical institutions (*n* = 4852), 605,126 cancer-related deaths were included in the study; of these, 17.79% were enrolled in palliative care in 2005, which increased to 62.51% by 2018, thus exhibiting an annually increasing tend (Figure 2). The main characteristics of the patients with terminal cancer are shown in Appendix A. Appendix A shows that of these, 39.61% of terminal cancer patients participated in palliative care before their deaths. Among them, the proportion of women participating in palliative care (43.52%) was higher than that of men (37.31%). In terms of age, those aged 55–64 had the highest participation rate (43.40%), and those aged 75–85 had the lowest participation rate (36.87%). The higher the education level of patients, the higher their rate of joining palliative care. Those with a college education or above had a joining rate of 44.57%, accounting for the highest rate. In addition, those with pancreatic cancer accounted for the highest proportion of patients (49.90%), while those with prostate cancer had the lowest proportion (34.37%).

With PSM, no significant differences were observed in sex, age, severity of comorbidities, and cancer type between the two groups (*p* > 0.05) (Appendix A).

### 3.2. Comparison of Differences in Emergency Care Utilization, CPR, ICU Admission, and Intubation among the Two Groups

The enrolled group had a lower rate of emergency care utilization (*p* < 0.05) compared to the unenrolled group at 1 week to 24 weeks before death (Table 1). Regarding the duration of enrollment in palliative care, the emergency care utilization rate for the enrolled group 1 week before death was 13.09%; with an increase in the duration of enrollment, the emergency care utilization rate increased markedly to 72.56% for enrolled patients 24 weeks before death. In addition, the CPR, ICU admission, and intubation rates of enrolled patients at 1 week to 24 weeks before death were significantly lower than of unenrolled patients (*p* < 0.05). Furthermore, the CPR utilization rate increased from 1.77% in enrolled patients 1 week before death to 3.74% in those enrolled 24 weeks before death; the ICU admission rate increased from 9.26% to 19.57%; and the intubation rate increased from 4.05% to 8.12% (Table 1). Overall, a longer duration of enrollment in palliative care before death was associated with greater emergency care utilization, CPR, ICU admission, and intubation, with the largest increase being in emergency care utilization (almost 5-fold), whereas the rates of CPR, ICU admission, and intubation increased 2-fold (Table 1).

Conditional logistic regression analyses showed that after controlling for relevant variables, the odds ratios for emergency care utilization, CPR, ICU admission, and intubation were lower (*p* < 0.05) for enrolled group at 1, 4, 8, 12, or 24 weeks before death compared to the unenrolled group (Table 2). The odds ratio for emergency care utilization among those enrolled at 1 week before death was 0.34-fold than that of those not enrolled (95% CI: 0.34–0.35). With an increase in the duration of enrollment, the odds ratio for emergency care utilization increased 0.68-fold for those enrolled at 24 weeks before death (95% CI: 0.65–0.72). The odds ratio for CPR utilization among those enrolled at 1 week before death was 0.13-fold that of those not enrolled (95% CI: 0.13–0.14) and 0.26-fold among those enrolled at 24 weeks before death (95% CI: 0.24–0.28). For ICU admission, the odds ratio was 0.27-fold among those enrolled at 1 week before death compared to that of those not enrolled (95% CI: 0.26–0.27) and was 0.40-fold among those enrolled at 24 weeks before death (95% CI: 0.39–0.43). For intubation, the odds ratio was 0.15-fold among those enrolled at 1 week before death compared to that of those not enrolled (95% CI: 0.15–0.16) and was 0.26-fold for those enrolled at 24 weeks before death (95% CI: 0.24–0.28) (Table 2). The odds ratio for emergency care utilization, CPR, ICU admission, and intubation for the enrolled group slowly decreased with an increase in the duration of palliative care before death compared to the unenrolled group.

### 3.3. Yearly Trends in the Differences in Palliative Care with Respect to Emergency Care Utilization, CPR, ICU Admission, and Intubation

Next, the utilization rates of the two groups were compared each year to observe the trends (Figure 3, Figure 4, Figure 5 and Figure 6 and Appendix A). Figure 3 shows that the ratio of emergency care utilization between the two groups had different yearly trends depending on the duration of palliative care before death. The ratio of those enrolled to those not enrolled at 1 week before death was level at 0.46–0.54-fold in 2005–2010, steadily decreasing to 0.40-fold in 2010–2018 (Appendix A). This is largely due to a large yearly increase in emergency care utilization among those not enrolled in 2005–2018. The ratio of utilization rates for those enrolled at 4, 8, and 12 weeks before death to those not enrolled increased steadily in 2005–2018, which was primarily due to a yearly increase in emergency care utilization among those not enrolled in 2005–2018. The ratio of utilization rates for those enrolled at 24 weeks before death to those not enrolled decreased slightly in 2005–2010 to a minimum in 2010 (0.83) and then gradually increased to 0.90-fold in 2011–2018, a trend that was largely due to a similar increase in emergency care utilization among both those enrolled and not enrolled in 2005–2018 (Appendix A).

The rate of CPR utilization among enrolled patients at 1, 4, and 8 weeks before death exhibited a steady decrease in 2005–2010 and an increase in 2010–2018 compared to that of those not enrolled (Figure 4). This is primarily due to the decrease in CPR utilization in both groups in 2005–2010, which was lower among the enrolled group compared to the unenrolled group (Appendix A). The rate among those enrolled at 12 and 24 weeks before death exhibited a steady increase in 2005–2010 and a decrease in 2010–2018 compared to that of those not enrolled; this is primarily due to higher CPR utilization among those enrolled in 2005–2010 (Appendix A).

The rate of intubation among those enrolled at 1, 4, 8, 12, and 24 weeks before death decreased over time in 2005–2010 compared to that of those not enrolled (Figure 5); however, the rate of intubation among the two groups increased over time in 2010–2018. However, the decrease in intubation rates was larger in the enrolled group; in 2010–2018, intubation rates decreased for the unenrolled group and increased for the enrolled group (Appendix A).

The rate of ICU admission for enrolled group at 1, 4, 8, 12, or 24 weeks before death decreased over time in 2005–2010 compared to that of those unenrolled group (Figure 6). In 2010–2018, the rates of ICU admission increased in both groups. This change is primarily due to changes in ICU admission among those enrolled in palliative care, which increased annually from 2010 to 2018 (Appendix A).

## 4. Discussion

The results of the present study show a yearly increase in the percentage of patients with terminal cancer receiving hospice and palliative care before death, from 17.79% in 2005 to 62.51% in 2018. Hospice and palliative care is increasingly available to such patients as an essential component of cancer treatment, and a higher percentage of cancer patients receive palliative care before death than patients who die of other causes [30].

Our findings show that the enrolled group had lower rates of emergency care utilization compared to the unenrolled group, similar to previous studies [17]. Patients with cancer seek emergency care at the end of life mostly owing to pain, respiratory distress, and gastrointestinal problems [9]. Hospice and palliative care can control symptoms such as respiratory distress, pain, and depression at the end of life [15], thus mitigating some of the need for emergency care. However, a few studies have found that patients with terminal cancer who received hospice and palliative care had a higher rate of emergency care utilization compared to those who did not receive it [31]. “Uncontrolled symptoms” in patients with terminal cancer are the primary reason for seeking emergency treatment. Therefore, patients who receive hospice and palliative care may make more emergency room visits at the end of life because hospice and palliative care is prioritized for patients with more severe disease. Herein, enrolled/not enrolled patients still had ICU admission rates of 9.3–36.3% at the time of death, similar to previous studies [32]. Hospice and palliative care consultation provides patients with terminal cancer with appropriate care; a timely and accurate understanding of the patient’s care preferences is a form of patient–physician communication that reduces the inappropriate utilization of medical resources, including ICU admission. Many previous reports have suggested that those who receive hospice and palliative care have lower rates of ICU utilization [33,34]. However, Liu et al. found that hospice and palliative care did not reduce ICU utilization [35]. The present study showed that ICU utilization was lower in the enrolled group at 1, 4, 8, 12, or 24 weeks before death regardless of when they were enrolled, compared to the unenrolled group.

The enrolled group also had significantly lower utilization of CPR (OR = 0.26) and intubation (OR = 0.26) (Table 2). Taiwan qualified the Hospice Palliative Care Act in 2000; therefore, patients receiving palliative care were counseled and given an opportunity to sign Do Not Resuscitate (DNR) orders. After the implementation of the palliative care consultation services (PCCS) program in Taiwan, the percentage of DNR consent also increased from 44.0% in 2006 to 80.0% in 2014 [36]. Advance directives record the wishes of patients regarding end-of-life (EOL) care. If patients complete advance directives, they are more likely to receive EOL care that aligns with their goals and wishes. Though Taiwan was the first Asian country to qualify something like the Hospice Palliative Care Act, there were only 570,000 people registered with a “pre-established intention for palliative care” as of November 2018, and death literacy among the Taiwanese public remains lacking. However, via health education by hospice and palliative care teams, terminally ill cancer patients receiving hospice and palliative care have more opportunities to encounter information regarding advance directives and choose DNR. This choice limits the utilization of futile emergency medical resources at the end of life, subsequently making the likelihood of receiving CPR and endotracheal intubation significantly lower than in patients who do not receive hospice and palliative care. Previous data from Taiwan showed that hospice and palliative care intervention reduced the odds ratio for tracheal intubation (OR = 0.71) in patients who died of cancer in 2001–2006 [23]. Herein, hospice and palliative care intervention reduced the odds ratio for intubation (OR = 0.26) in patients who died of cancer in 2005–2018 even lower than in previous studies, suggesting that hospice and palliative care significantly reduces futile medical interventions in patients with terminal cancer using relevant health education.

With respect to the effect of the “duration of enrollment in palliative care” on emergency care utilization, CPR, ICU admission, and intubation of patients with terminal cancer, we found that the difference in adverse care indicators between the two groups slowly decreased with an increasing duration of enrollment in palliative care. Previous studies have shown that earlier enrollment in hospice and palliative care results in less utilization of healthcare resources; Hui et al. showed that patients enrolled in hospice and palliative care at early stages (90 days before death) had significantly lower emergency room visits (39% for early enrollees vs. 68% for late enrollees) and ICU admission (6% for early enrollees vs. 11% for late enrollees) within 1 month before death than those enrolled in palliative at later stages (within 90 days before death) [25]. Present findings differ from those of previous studies, mainly because the latter have used a 1-month period before death as the criterion for evaluating the utilization of healthcare resources, whereas the present study considered the time frame from the start of palliative care to death. Later enrollment in hospice and palliative care is often associated with more severe illness and a lower likelihood that the patient and family members will opt for aggressive treatments [31]. Therefore, later enrollment in hospice and palliative care decreases the likelihood of medically futile measures such as CPR, ICU admission, and intubation, as well as a decreased probability of the improper use of healthcare resources compared to those enrolled in palliative care at early stages (24 weeks before death).

In 2005–2018, a yearly increase in emergency care utilization was observed in both groups. The NHI system of Taiwan is highly accessible and convenient, and patients with terminal cancer are exempted from some of the burdens of medical treatment under the system. Studies have shown that patients with major injuries and illnesses tend to have more emergency room visits and higher emergency room costs due to atypical medical symptoms [37]. Patients with terminal cancer have fewer financial considerations for using medical treatment (some costs are waived); therefore, they often seek emergency medical treatment due to atypical symptoms and to reduce discomfort, thus increasing the rate of emergency care utilization [37,38]. Additionally, enrolled patients have better control of their disease due to proper health education and have an inherently lower probability of requiring emergency care compared to those not enrolled. The strategies of hospice and palliative care aimed at alleviating pain may encompass tumor reduction treatments or the administration of tailored pharmacological pain control measures. These interventions constitute choices within the realm of hospice and palliative care, dedicated to fostering physical comfort and pain relief for terminal cancer patients, and thus may increase emergency care utilization.

The utilization rates of CPR, intubation, and ICU admission—all life-sustaining procedures rather than life-prolonging treatments—in enrolled patients exhibited annual decreases in 2005–2010 and annual increases in 2010–2018. In Taiwan, hospice and palliative care at home for terminal cancer patients was covered under universal health insurance in 1996. The decreased utilization in 2005–2010 can be attributed to the passage of the Hospice Palliative Care Act in 2000, which permitted the abandonment of CPR as a legal medical practice, thereby legitimizing palliative care for patients with terminal cancer. Although the PCCS program was implemented in 2005 to reduce futile treatments (CPR, intubation, and ICU admission) in hospice and palliative care for terminal cancer, the utilization of CPR, intubation, and ICU admission increased in 2010–2018; this may be attributed to the shift from a pilot program to formal inclusion of palliative care in healthcare benefits on 1 September 2009, and the inclusion of eight major categories of non-cancer illnesses in the scope of palliative care. This expanded the target population of palliative care and resulted in a shortage of professional nursing manpower, with inadequate health education and death literacy among patients with terminal cancer and their caregivers, leading to increased rates of utilization of medically futile measures. With advances in various targeted anticancer drugs, the public has come to expect new drugs to be effective and the treatment of end-stage cancer has become more aggressive [39]. Moreover, the inclusion of anticancer drugs in the NHI system has reduced the financial burden of patients with cancer, another possible explanation for the increasing use of the emergency room, chemotherapy (including experimental targeted drugs), and hospitalization, leading to increased rates of CPR, intubation, and ICU admission each year.

Although the Palliative and Hospice Care Act allows individuals to opt for refusal of cardiopulmonary resuscitation or life-sustaining treatments (Do Not Resuscitate (DNR)) at the end of life, practical implementation still faces numerous obstacles, such as physicians’ awareness of the relevant laws and emergency resuscitations conducted by physicians at the request of family, even when the patient has a clear DNR directive, due to the family’s lack of understanding of the patient’s condition. It is therefore essential to proactively educate both patients and their families to ensure a comprehensive understanding of hospice and palliative care and DNR directives, aiming to shorten the duration of suffering and uphold the dignity of terminal patients. Further, this study found that as the time from enrollment in hospice and palliative care to death increases, the differences in the likelihood of emergency, ICU admission, intubation, and CPR usage between participants and non-participants of palliative care gradually diminish. Patients who are enrolled in hospice and palliative care longer before death may neglect the essence of hospice and palliative care, leading to futile medical interventions during emergencies. Thus, in addition to actively promoting DNR directives among hospice and palliative care patients, it is crucial to enhance the documentation of DNR preferences in the National Health Insurance system. This will allow physicians to instantly access pre-established patient wishes, reducing the use of futile medical interventions.

### Limitations of the Study

Because this study analyzed secondary data, it was not possible to obtain the personal characteristics of the primary caregivers of patients with terminal cancer or their perceptions of palliative care; it was only possible to control for healthcare providers (healthcare facility and attending physician) and patient characteristics. Furthermore, it was not possible to investigate the influence of the primary caregiver on whether a patient is enrolled in palliative care. In addition, the study was limited by the inability to identify the stage of cancer at the time of enrollment in palliative care, and the inability to know whether any patients were enrolled in hospice and palliative care for non-terminal cancers. The hospice and palliative care resources and implementation plans in Taiwan may differ from those in other countries, thereby making the trends in emergency medical resource utilization by terminal cancer patients unique. Consequently, the extrapolation of the findings from this study to other countries might be limited.

## 5. Conclusions

The results of the present study show a yearly increase in the percentage of patients with terminal cancer receiving hospice and palliative care before death, from 17.79% in 2005 to 62.51% in 2018. Terminal cancer patients enrolled in palliative care had lower rates of emergency care utilization, CPR, ICU admission, and intubation than those not enrolled, regardless of the duration between enrollment in palliative care and death. However, with an increasing duration of enrollment in hospice and palliative care before death, the difference in emergency care utilization, ICU admission, intubation, and CPR gradually decreased. The PCCS was piloted in 2005, and patients enrolled in palliative care in 2005–2010 exhibited decreasing rates of CPR, intubation, and ICU admission. However, after the full inclusion of hospice and palliative care in healthcare coverage in 2010, the rates of aggressive life-sustaining interventions such as emergency care, CPR, intubation, and ICU admission increased slightly each year in 2005–2010, even among those enrolled. This finding warrants the attention of health policymakers when developing policies related to palliative care.

## Figures and Tables

**Figure 1 healthcare-11-02907-f001:**
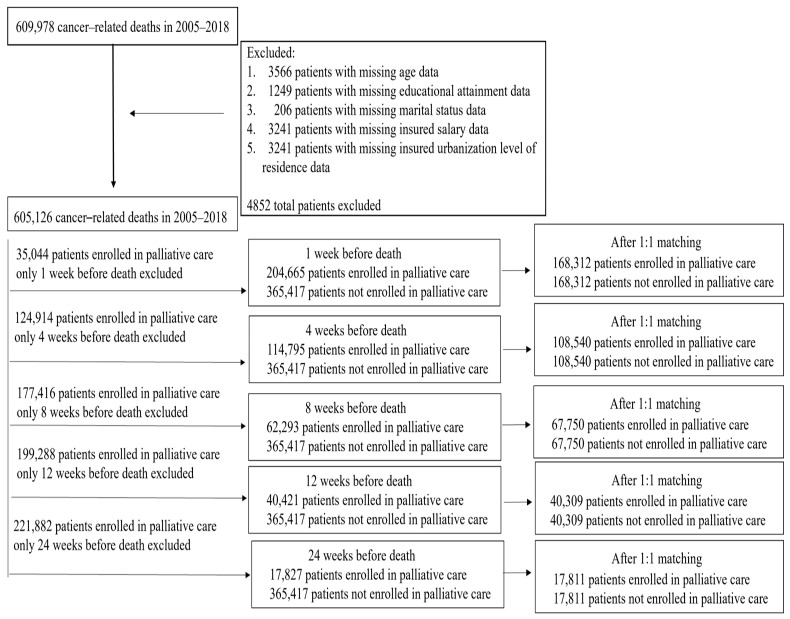
Flowchart of the selection process.

**Figure 2 healthcare-11-02907-f002:**
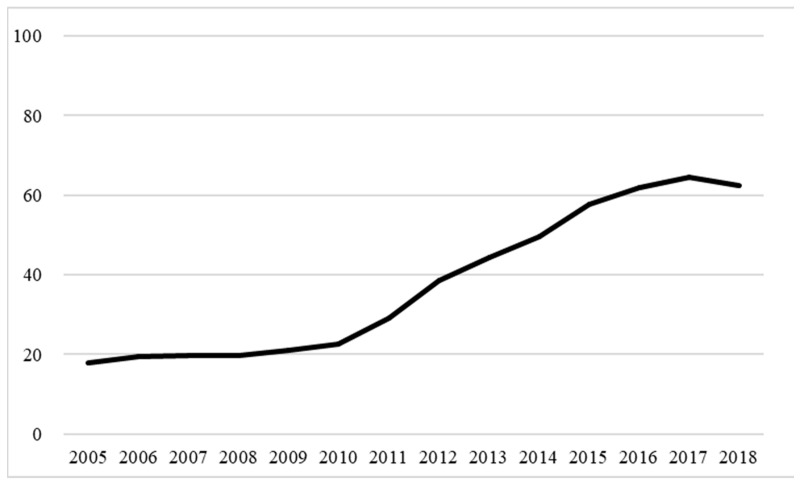
Trends in the percentage of terminal cancer patients enrolled in palliative care before death, 2005–2018.

**Figure 3 healthcare-11-02907-f003:**
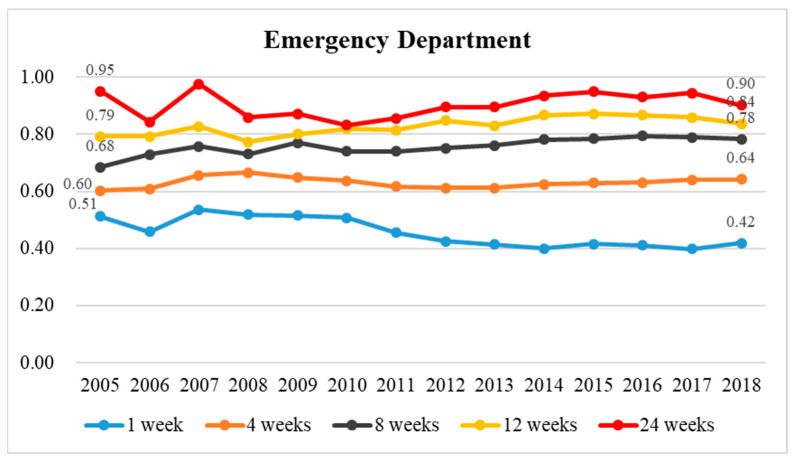
The historical trend in the ratio of emergency department utilization among terminal cancer patients whether in palliative care or not between 2005 and 2018 and the ratio of emergency department utilization rates for those enrolled to those not enrolled in terminal cancer patients at 1, 4, 8, 12, and 24 weeks before death.

**Figure 4 healthcare-11-02907-f004:**
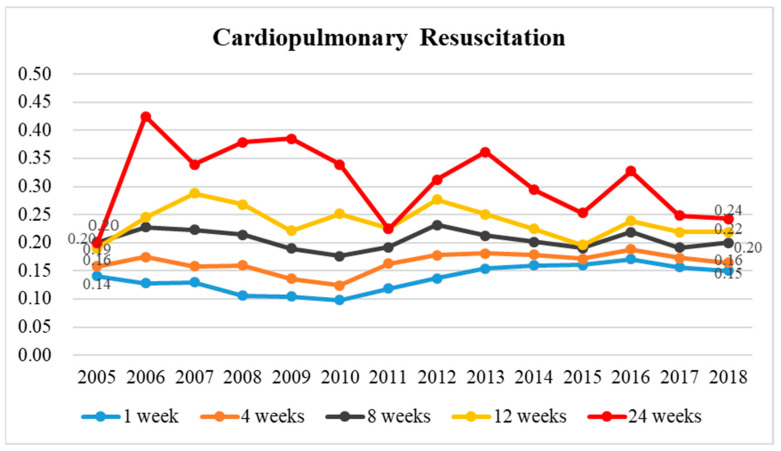
The historical trend in the ratio of cardiopulmonary resuscitation among terminal cancer patients depending on whether they were in palliative care or not between 2005 and 2018 and the ratio of cardiopulmonary resuscitation utilization rates for those enrolled to those not enrolled in terminal cancer patients at 1, 4, 8, 12, and 24 weeks before death.

**Figure 5 healthcare-11-02907-f005:**
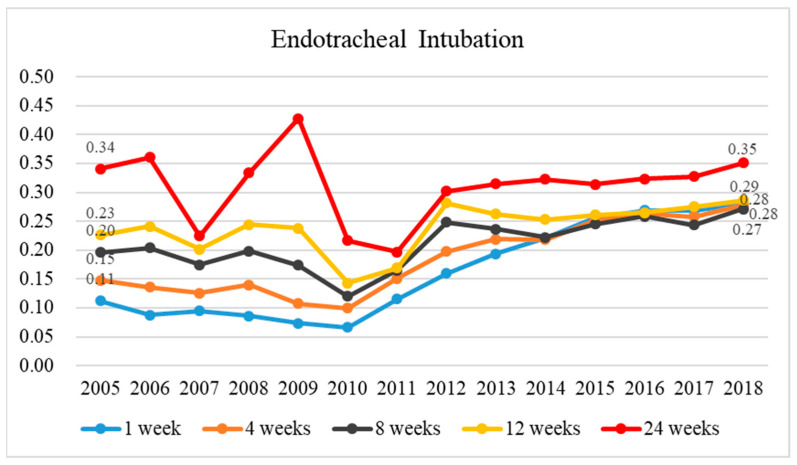
The historical trend in the ratio of endotracheal intubation utilization among terminal cancer patients depending on whether they were in palliative care or not between 2005 and 2018 and the ratio of endotracheal intubation utilization rates for those enrolled to those not enrolled in terminal cancer patients at 1, 4, 8, 12, and 24 weeks before death.

**Figure 6 healthcare-11-02907-f006:**
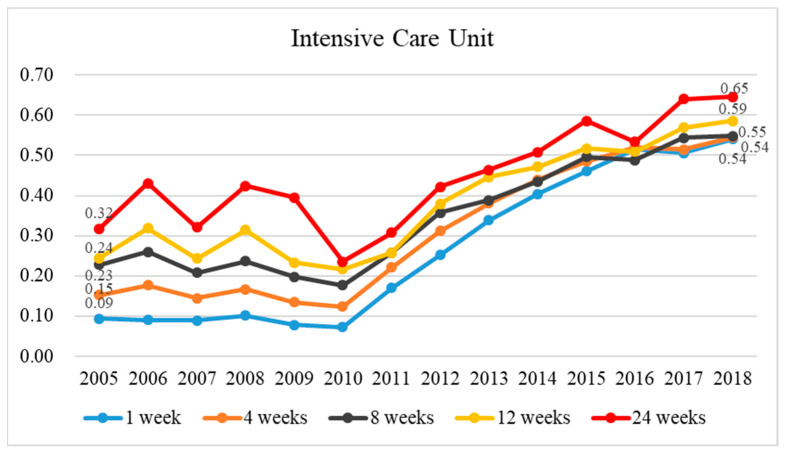
The historical trend in the ratio of ICU admission among terminal cancer patients depending on whether they were in palliative care or not between 2005 and 2018 and the ratio of ICU admission utilization rates for those enrolled to those not enrolled in terminal cancer patients at 1, 4, 8, 12, and 24 weeks before death.

**Table 1 healthcare-11-02907-t001:** Differences in emergency care utilization, CPR, ICU admission, and intubation in enrolled vs. unenrolled patients with terminal cancer in palliative care at 1, 4, 8, 12, or 24 weeks before death.

	1 Week before Death					4 Weeks before Death					8 Weeks before Death					12 Weeks before Death					24 Weeks before Death				
			Utilization of Healthcare Resources		χ^2^			Utilization of Healthcare Resources		χ^2^			Utilization of Healthcare Resources		χ^2^			Utilization of Healthcare Resources		χ^2^			Utilization of Healthcare Resources		χ^2^
Variable	N	%	n_1_	%	*p*-Value	N	%	n_1_	%	*p*-Value	N	%	n_1_	%	*p*-Value	N	%	n_1_	%	*p*-Value	N	%	n_1_	%	*p*-Value
Emergency care total	336,624	100.00	73,299	21.77		217,080	100.00	98,425	45.34		123,500	100.00	73,656	59.64		80,618	100.00	53,959	66.93		35,622	100.00	27,006	75.81	
Palliative care					<0.001					<0.001					<0.001					<0.001					<0.001
Not enrolled	168,312	50.00	51,264	30.46		108,540	50.00	60,369	55.62		61,750	50.00	41,523	67.24		40,309	50.00	29,216	72.48		17,811	50.00	14,082	79.06	
Enrolled	168,312	50.00	22,035	13.09		108,540	50.00	38,056	35.06		61,750	50.00	32,133	52.04		40,309	50.00	24,743	61.38		17,811	50.00	12,924	72.56	
CPR total	336,624	100.00	23,434	6.96		217,080	100.00	16,175	7.45		123,500	100.00	9640	7.81		80,618	100.00	6446	8.00		35,622	100.00	3005	8.44	
Palliative care					<0.001					<0.001					<0.001					<0.001					<0.001
Not enrolled	168,312	50.00	20,458	12.15		108,540	50.00	13,811	12.72		61,750	50.00	8006	12.97		40,309	50.00	5244	13.01		17,811	50.00	2338	13.13	
Enrolled	168,312	50.00	2976	1.77		108,540	50.00	2364	2.18		61,750	50.00	1634	2.65		40,309	50.00	1202	2.98		17,811	50.00	667	3.74	
ICU total	336,624	100.00	61,166	18.17		217,080	100.00	44,118	20.32		123,500	100.00	27,827	22.53		80,618	100.00	19,561	24.26		35,622	100.00	9967	27.98	
Palliative care					<0.001					<0.001					<0.001					<0.001					<0.001
Not enrolled	168,312	50.00	45,574	27.08		108,540	50.00	31,491	29.01		61,750	50.00	19,431	31.47		40,309	50.00	13,308	33.01		17,811	50.00	6482	36.39	
Enrolled	168,312	50.00	15,592	9.26		108,540	50.00	12,627	11.63		61,750	50.00	8396	13.60		40,309	50.00	6253	15.51		17,811	50.00	3485	19.57	
Intubation total	336,624	100.00	42,203	12.54		217,080	100.00	29,858	13.75		123,500	100.00	18,266	14.79		80,618	100.00	12,467	15.46		35,622	100.00	5958	16.73	
Palliative care					<0.001					<0.001					<0.001					<0.001					<0.001
Not enrolled	168,312	50.00	35,380	21.02		108,540	50.00	24,530	22.60		61,750	50.00	14,821	24.00		40,309	50.00	9918	24.60		17,811	50.00	4512	25.33	
Enrolled	168,312	50.00	6823	4.05		108,540	50.00	5328	4.91		61,750	50.00	3445	5.58		40,309	50.00	2549	6.32		17,811	50.00	1446	8.12	

CPR, cardiopulmonary resuscitation; ICU, intensive care unit.

**Table 2 healthcare-11-02907-t002:** Effects on emergency care utilization, CPR, ICU admission, and intubation in patients with terminal cancer that were enrolled or not enrolled in palliative care at 1, 4, 8, 12, or 24 weeks before death and their associated factors *.

	1 Week before Death				4 Week before Death				8 Week before Death				12 Week before Death				24 Week before Death			
Variable	OR	95% CI		*p*-Value	OR	95% CI		*p*-Value	OR	95% CI		*p*-Value	OR	95% CI		*p*-Value	OR	95% CI		*p*-Value
Emergency																				
Palliative care																				
Enrolled vs. no (ref)	0.34	0.34–0.35		<0.001	0.41	0.41–0.42		<0.001	0.50	0.49–0.50		<0.001	0.57	0.56–0.59		<0.001	0.68	0.65–0.72		<0.001
CPR																				
Palliative care																				
Enrolled vs. no (ref)	0.13	0.13–0.14		<0.001	0.15	0.15–0.16		<0.001	0.18	0.17–0.19		<0.001	0.21	0.19–0.22		<0.001	0.26	0.24–0.28		<0.001
ICU																				
Palliative care																				
Enrolled vs. no (ref)	0.27	0.26–0.27		<0.001	0.31	0.31–0.32		<0.001	0.33	0.32–0.34		<0.001	0.36	0.34–0.37		<0.001	0.40	0.39–0.43		<0.001
Intubation																				
Palliative care																				
Enrolled vs. no (ref)	0.15	0.15–0.16		<0.001	0.17	0.17–0.18		<0.001	0.18	0.18–0.19		<0.001	0.20	0.20–0.21		<0.001	0.26	0.24–0.28		<0.001

* All models have been controlled for patient sex, age, educational attainment, marital status, monthly salary, urbanization level of residence, severity of comorbidities, type of cancer, primary care provider ownership and level of care, and year of death. CPR, cardiopulmonary resuscitation; ICU, intensive care unit.

## Data Availability

Data are available from the Health and Welfare Data Science Center of the Ministry of Health and Welfare (MOHW), Taiwan. All interested researchers can apply to use the database managed by the MOHW. Due to legal restrictions imposed by the Taiwanese government related to the Personal Information Protection Act, the database cannot be made publicly available. Any raw data are not allowed to extracted from the Health and Welfare Data Science Center. The restrictions prohibited the authors from making the minimal data set publicly available.

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
