# Peer review of "Effectiveness of Palliative Care before Death in Reducing Emergency Care Utilization for Patients with Terminal Cancer and Trends in the Utilization of Palliative Care from 2005–2018"

_healthcare, 2023, doi:10.3390/healthcare11212907_

Round 1
Reviewer 1 Report
Comments and Suggestions for Authors
This study presents a two-fold objective; first, to compare differences in the use of several healthcare services among patients with terminal cancer (enrolled and not enrolled in palliative care) at 1, 4, 8, 12, and 24 weeks 78 before death. ; and, second, to understand trends in the utilization of palliative care by these patients before death from 2005–2018.
The second objective is not well reflected on the title.
The abstract is a bit confusing, providing a relevant amount of results that would require a better presentation to avoid confusion on the reader. Within the abstract, the type of study performed should be stated, as well as the number of cases of patients used. It is not clear what types of patients the study is focused on: only adults? The term CPR should be defined.
In the methodology I miss information on the main characteristics of the patients: mean age, general population or only adults, sex, type of cancer, etc.
At lines 42-43 it is mentioned that <<emergency care utilization at the end of life is considered an important indicator of end-of-life cancer treatment and quality of life>>. In which sense is the QoL an important indicator? I guess, authors mean a deteriorated QoL, but this should be clarified.
At the Results section, authors mention that at Figures 3-6 the trend of the use of several services among terminal cancer patients whether to join palliative care or not between 2005 and 2018 are shown. However, I am only able to detect the trends of patients enrolled in palliative at different moments (at 1, 4, 8, 12, and 24 weeks before death), but not the trend from those patient not involved in palliative care pathways. Further clarification should be provided.
At lines 332-336 it is stated the following <<Therefore, later enrolment in palliative care decreases the likelihood of medically futile measures such as CPR, ICU admission, and intubation as well as a decreased probability of improper use of healthcare resources compared to those enrolled in palliative care at early stages (24 weeks before death).>> These results and affirmations apparently are against the wide evidence it is already available on the benefits of early approaches to palliative for terminally ill cancer patients. In this sense, how do authors interpret these results?
At lines 351-372 trends on the use of different services depending on the state of palliative care resources and plans around its implementation are presented. Do authors know if these trends are also showed in other contexts or countries? Or is this something unique at this specific context of study?
The Discussion would need more details on the practical implications of the results obtained in this study.
Reviewer 2 Report
Comments and Suggestions for Authors
Thank you for the opportunity to review this manuscript. The researchers did a good job here and the topic is relevant, as well as a solid method of analysis.
Here are a couple suggestions to improve the manuscript. Interested to hear the authors' feedback on these suggestions:
- the independent variables (CPR, ICU admission, and intubation of patients) should be discussed further, specifically addressing multicollinearity. Please discuss (comment) about how patients undergoing CPR, often are also intubated, often in the ER, and/or frequently admitted to the ICU (based on acuity levels).
- figures 3, 4, 5, and 6 - why are they not displayed to show collective results of those in PC programs and those not in PC programs? Seems like total sample reporting here, versus broken-out by the dependent variable. Would this not be more appropriate to show?
- This study provides a high-level logistic regression of PC program participation. However, based on the independent variables, as associated with end-of-life high intensity lifesaving care, please discuss more on implied consent and the advanced directive topic. While addressed - those in a PC program may receive more information on an advanced directive, limiting emergency resources at the end-of-life, versus those without an advanced directive and therefore implied consent taking over in emergency situations (significant resources utilized). This would be a good area for future study to suggest as well.
Reviewer 3 Report
Comments and Suggestions for Authors
Tsai et al have submitted a manuscript entitled ’Effectiveness of duration of palliative care before death in reducing emergency care utilization for patients with terminal cancer’.
Clinical context of the study is very relevant. Study includes retrospective analysis of patients who died of cancer form 2005 till 2018 and describe changes in palliative care organization more than over decade in Taiwan.
I have some comments.
Effectiveness of duration of palliative care is somehow misleading. Patients did not start palliative care at the time of diagnoses, instead data was analyzed retrospectively from the date of death. Natural course of advanced cancer is worsening of general health and symptom control over time with progressing cancer. Hence, it is implicit that palliative care becomes more and more relevant closer to the death.
Last sentence of abstract: Patients with terminal cancer enrolled in palliative care consume fewer medical resources before death than unenrolled patients, however, the difference decreases with longer times of enrollment.
This is not longer time of enrollment, rather longer time before death (or further away from death).
The Authors aimed to describe effectiveness of duration of palliative care.
Seems that major factor impacting utilization of emergency care is determined by year patient died.
Abstract, line 24: Since inclusion of palliative care in NHI (2010 onwards), its utilization increased slightly each year.
The Authors have not been able to demonstrate that longer before death patient is enrolled in palliative care has impact on emergency care utilization. Figure 3-6 – no change in utilization of ICU, ETI, CPR were seen regardless of duration of palliative care from 4 till 24 weeks before death. Duration of palliative care impacted only utilization of emergency department visits, eg enrolment into palliative care reduced emergency department visits, particularly closer to the death.
I think main findings are described well in tables and very clearly present changes in emergency care before death in patients enrolled and not enrolled into palliative care over time.
Main conclusion: ‘after full inclusion of palliative care in health care coverage in 2010, the rates of aggressive life-sustaining interventions such as emergency care, CPR, intubation, and ICU admission increased slightly each year in 2005–2010 even among those enrolled.’ could also be presented in Abstract, and highlights unnecessary use of health care resource. Also, ‘The present study found an increase in emergency care utilization over the study period, suggesting that palliative care is primarily a way for patients with cancer to receive active care at the end of life, rather than simply giving up on extensive treatment’.
Generally emergency department visits by patients with terminal cancer should be avoid since it acute care, care out of normal working hours, is burden related to high costs. Active care at the end of life is not needed.
These are main findings and maybe the title and abstract could better reflect this. Maybe Authors could also explain why palliative care is primarily a way for patients with cancer to receive active care at the end of life (eg contradicts with general understanding of terminal care) (do hospice or palliative care hospitals have their own E&A departments?).
